# Robustness of Rotation-Equivariant Networks to Adversarial Perturbations

**Béranger Dumont, Simona Maggio & Pablo Montalvo** [*]
Rakuten Institute of Technology Paris
`{beranger.dumont,simona.maggio}@rakuten.com`,
`pablo.montalvo.leroux@gmail.com`

## Abstract

Deep neural networks have been shown to be vulnerable to adversarial examples: very small perturbations of the input having a dramatic impact on the predictions. A wealth of adversarial attacks and distance metrics to quantify the similarity between natural and adversarial images have been proposed, recently enlarging the scope of adversarial examples with geometric transformations beyond pixel-wise attacks. In this context, we investigate the robustness to adversarial attacks of new Convolutional Neural Network architectures providing equivariance to rotations. We found that rotation-equivariant networks are significantly less vulnerable to geometric-based attacks than regular networks on the MNIST, CIFAR-10, and ImageNet datasets.

## 1 Introduction

Deep learning provided significant breakthroughs in machine learning, and Convolutional Neural Networks (CNNs) are now being used routinely for computer vision tasks. However, theoretical as well as practical concerns remain, a prime example being adversarial attacks: very small perturbations of the input causing catastrophic changes in the predictions of the network. Since adversarial examples were first noticed (Szegedy et al., 2014; Goodfellow et al., 2015), the vast majority of the studies has focused on pixel-wise attacks for the $L_0$, $L_2$, or $L_\infty$ distance metrics on images (for a review, see, *e.g.*, Akhtar & Mian (2018)). Very recently the notion of adversarial examples on images was refined to account for the human perception (Luo et al., 2018), and extended to rigid geometric transformations (Fawzi & Frossard, 2015; Kanbak et al., 2017; Engstrom et al., 2017) and local geometric distortions (Xiao et al., 2018).

In an independent effort, several groups strived to extend the symmetry properties of CNNs beyond equivariance to translations. The most natural next step is to equip the models with invariance or equivariance to rotations, so that a rotation of the input image will either leave the feature maps unchanged or will rotate them accordingly. Most of the existing approaches fall into two categories: the ones based on the rotation of the input images (Jaderberg et al., 2015; Laptev et al., 2016; Henriques & Vedaldi, 2017; Esteves et al., 2018), and the ones based on constraints on the structure of the filters (Cohen & Welling, 2016; 2017; Dieleman et al., 2016; Worrall et al., 2017; Zhou et al., 2017; Gonzalez et al., 2017; Li et al., 2017; Weiler et al., 2017).

We assess the robustness to adversarial examples of four distinct rotation-equivariant CNN architectures: Group Equivariant Convolutional Neural Networks (G-CNNs, Cohen & Welling (2016)), Harmonic Networks (H-Nets, Worrall et al. (2017)), Deep Rotation Equivariant Networks (DRENs, Li et al. (2017)) and Oriented Response Networks (ORNs, Zhou et al. (2017)). First, G-CNNs provide equivariance to 90 degrees rotations and mirror reflections by redefining the convolution operator over symmetry groups. Second, H-Nets obtain equivariance to rotations of arbitrary angles using complex-valued filters constrained to the family of circular harmonics. Third, DRENs exploit the cyclic properties of 90 degrees rotations, using rotated filters instead of rotating feature maps, to obtain a deep representation of rotation equivariance. Finally, ORNs obtain invariance to rotations using filters that actively rotate during convolution.

---

[*]PM is currently Visiting Scientist at the Rakuten Institute of Technology Paris.

All of these models have achieved (or defined a new) state-of-the-art performance on the rotated MNIST dataset (Larochelle et al., 2007), while G-CNNs, DRENs and ORNs have proven to be competitive on the CIFAR-10 dataset (Krizhevsky & Hinton, 2009) as well. All networks provide patch-wise rotation equivariance, thus might be more robust to local geometric transformations. We present our experimental setup in Section 2, our results in Section 3, and we conclude in Section 4.

## 2   EXPERIMENTAL SETUP

We train rotation-equivariant CNNs, as well as regular CNNs for comparison, on the MNIST, CIFAR-10 and ImageNet (Deng et al., 2009) datasets. We then evaluate those models against adversarial examples generated from the corresponding datasets. In the case of MNIST, we first consider a model with seven $5 \times 5$ convolutional layers, following closely the definition and training procedure from Worrall et al. (2017); Cohen & Welling (2016). We suppress as much as possible architecture-related specificities to compare the models on equal grounds, and consider the same number of parameters of about $34$k for all models.

As for CIFAR-10, we consider a 44-layer residual network, ResNet-44, and its corresponding G-CNN version which we call G-ResNet. We follow the exact same procedure as Cohen & Welling (2016), and adjust the number of filters per layer in order to obtain the same number of parameters for both models (2.6M). We were not able to achieve a competitive accuracy with H-Nets.

In the case of ImageNet, to the best of our knowledge only ORNs have public results. A pre-trained residual model with 18 convolutional layers, OR-ResNet-18, was provided by the authors. We compare its performances to the standard Torch implementation of ResNet-18. Both models have a total of $1.4$M parameters. For the training and validation of all networks, we have used publicly available implementations.[1]

We consider recently proposed adversarial attacks based on geometric transformations. First, rigid geometric transformations (global rotations and translations), following closely the procedure of Engstrom et al. (2017), but considering a different range: translations of $\pm 3$ pixels on both axes, and rotation of $\pm 10$ degrees by step of one degree for all datasets. In the case of CIFAR-10, we have compared zero and edge paddings, and found no significant difference. For ImageNet, we perform rotation and translation of the $256 \times 256$ images before cropping to $224 \times 224$.

Next, we consider spatially transformed adversarial examples (stAdv, Xiao et al. (2018)), which are white-box targeted attacks based on Spatial Transformer Networks (Jaderberg et al., 2015). The generated adversarial examples are designed to be misclassified while keeping the spatial transformation distance low. To balance adversarial and flow losses, as defined in Xiao et al. (2018), we take $\tau = 0.10$. For each sample we generate a set of stAdv attacks, taking each possible wrong label as a target.

We also wanted to assess the robustness of rotation-equivariant networks to popular pixel-wise attacks on the $L_p$ norm, and to that end we considered the Fast Gradient Sign (Goodfellow et al., 2015) and DeepFool (Moosavi-Dezfooli et al., 2017) attacks. They managed to completely fool the classifiers in almost all our cases, showing that there is no significant added robustness to these attacks from rotation-equivariant architectures.

## 3   RESULTS

For every model and dataset, we report results on the single-crop error rate on the natural test set (*i.e.*, with no adversarial perturbation). The robustness to adversarial attacks is quantified with the *attack success rate* (ASR): the average fraction of attacks fooling a classifier, for a given type of attack. In the computation of the ASR, we exclude samples from the test set which are misclassified even if no perturbation is applied.

For the models trained on MNIST, we obtained a test error of less than $0.7\%$ for all cases. The results are shown in Table 1 (top). We find that H-Nets perform worse than the baseline on rotation-based

---

[1] See `tscohen/GrouPy`, `tscohen/gconv_experiments` (G-CNN), `microljy/DREN` (DREN), `deworrall92/harmonicConvolutions` (H-Net), `ZhouYanzhao/ORN` (ORN), and `facebook/fb.resnet.torch` (ResNet baseline on ImageNet) on `GitHub.com`.

Table 1: Error on the natural test set and attack success rate (ASR) on the MNIST (top), CIFAR-10 (bottom left), and ImageNet (bottom right) datasets. R+T indicates that the attacks are made from a combination of rotations and translations, while R and T are for rotation- and translation-only attacks, respectively. stAdv are spatially transformed adversarial attacks.

| MNIST | | CNN | H-Net | G-CNN | DREN |
|---|---|---|---|---|---|
| Error | | 0.68% | 0.69% | 0.55% | 0.51% |
| ASR | R+T | 6.2% | 20.5% | 3.4% | **1.3%** |
| | R | 1.2% | 4.8% | 0.9% | **0.5%** |
| | T | 2.6% | 17.6% | 1.4% | **0.5%** |
| | stAdv | 92.6% | **77.7%** | 87.1% | 91.9% |

| CIFAR-10 | | ResNet | G-ResNet | | ImageNet | | ResNet | OR-ResNet |
|---|---|---|---|---|---|---|---|---|
| Error | | 8.9% | 6.1% | | Error | | 30.6% | 28.9% |
| ASR | R+T | 88.4% | **70.1%** | | ASR | R+T | 6.8% | **5.7%** |
| | R | 77.9% | **52.1%** | | | R | 6.5% | **5.4%** |
| | T | 88.8% | **66.5%** | | | T | **3.4%** | 4.3% |
| | stAdv | 83.4% | **83.1%** | | | | | |

attacks, which is consistent with the observations of Li et al. (2017). However, they tend to be more accurate against spatially transformed adversarial attacks, which could show that H-nets are robust to local deformations due to their equivariance to local rotations of arbitrary angles. G-CNNs are shown to be more robust to rigid geometric transformations than the CNN baseline, which shows that the learned representations are useful against these types of attacks. Overall DRENs perform best against rotation-based attacks, with 0.5% ASR for pure rotations. The superiority of DRENs might be coming from their intensive use of each rotated filter in the isotonic layers.

Second, on the CIFAR-10 dataset, we trained a ResNet-44 and obtained a test error of 8.9%, and trained a rotation-equivariant G-ResNet and obtained a test error of 6.1%. Results are shown in Table 1 (bottom left). All attacks are much more successful than on MNIST since CIFAR-10 images have a much richer content while having a comparable size. Overall, the G-ResNet achieves far lesser vulnerability to attacks by rotation and translation than the regular ResNet, and is marginally better against the stAdv attacks. For pure rotations, we have a drop in the ASR of 33%.

Finally, on the larger ImageNet dataset, we used a pre-trained ResNet-18 (resp. OR-ResNet-18) that achieves 30.6% error (resp. 28.9%) on the test set. We tested rotation and translation-based attacks on those models. Results are shown in Table 1 (bottom right). We can see that ORNs bring an improvement against rotation attacks, and are slightly worse than a regular ResNet with translation attacks. For combined rotations and translations, there is a drop in the ASR of 19%.

## 4 CONCLUSIONS

The investigation of changes in the architecture of CNNs as a defense against adversarial examples is a relatively unexplored area. In order to evaluate the robustness of rotation-equivariant networks to adversarial examples, we conducted experiments on four different types of CNN architectures: G-CNNs, H-Nets, DRENs, and ORNs, on three different datasets: MNIST, CIFAR-10, and ImageNet. The networks which are equivariant to rotations by discrete angles were found to be significantly more robust to attacks based on small translations and rotations and, more marginally, to attacks based on local geometric distortions. We hope that this work will serve as a motivation for further studies on CNNs with extended symmetries, as well as for exploring the interplay between the natural robustness of the rotation-equivariant architectures to geometric-based adversarial attacks and other mechanisms of defense against adversarial examples.

### ACKNOWLEDGMENTS

We thank Andrés Hoyos-Idrobo for the useful discussions and Laurent Ach for his support.

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

## APPENDIX A    ANALYSIS OF THE ATTACK SUCCESS RATE

We also analyzed the distribution of the attack success rate (ASR) on the test set for the three datasets: MNIST, CIFAR-10, and ImageNet. Results are shown in Fig. 1. We can see that the attack produces different distributions for H-Nets and regular CNNs, but G-CNNs and DRENs are fairly similar to the regular one. In the case of CIFAR-10, there is a strong asymmetry in the regular case that is less pronounced in the G-ResNet case. As for ImageNet, both distributions are smoothly falling but as a much lower rate than for MNIST. Further investigation is needed to fully understand these results.

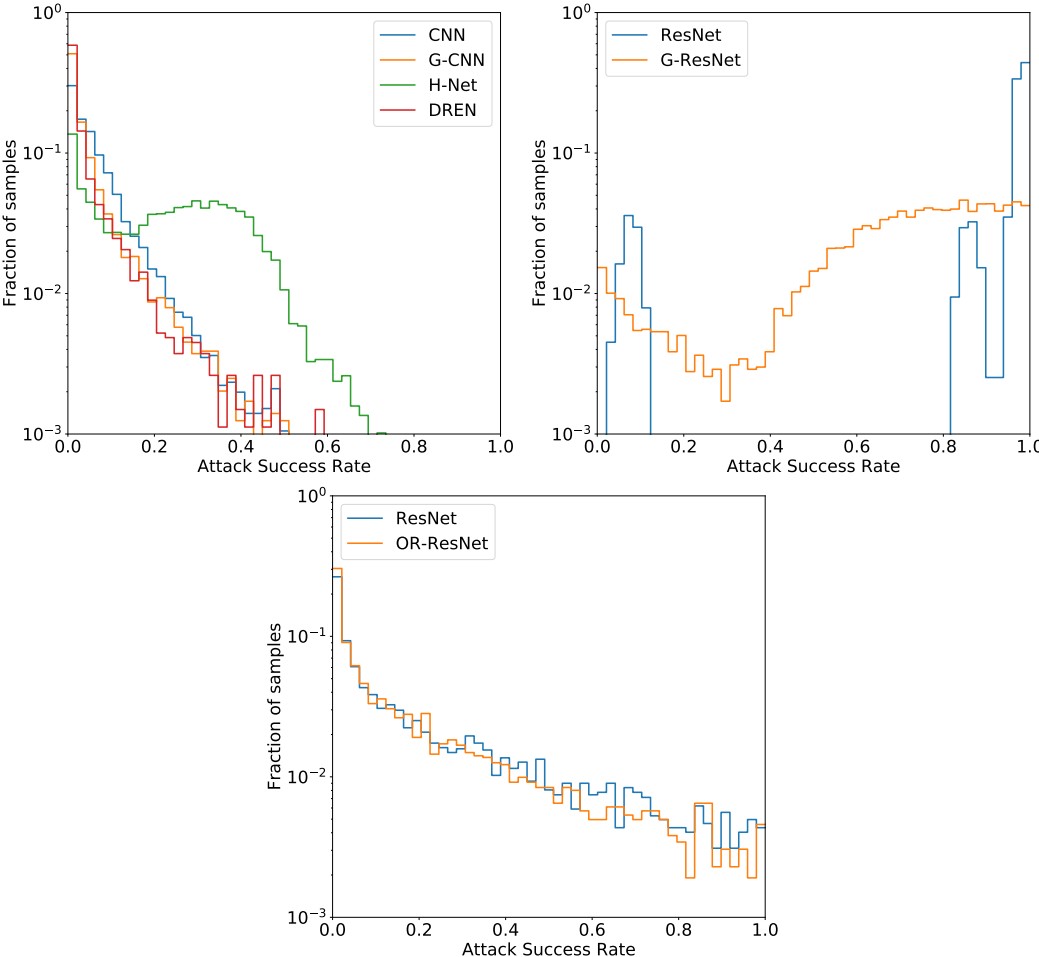

Figure 1: Distributions of the attack success rate on the test set for MNIST (top left), CIFAR-10 (top right), and ImageNet (bottom).

Finally, we checked the correlation coefficient between the ASR and the absolute value of the rotation angle. In the case of MNIST, all rotation-equivariant networks are less sensitive to the rotation angle than the regular CNN. We observe the opposite behavior in the case of the residual networks we consider for CIFAR-10, while there is no significant difference for the two models tested on ImageNet.

Table 2: Correlation coefficient between the attack success rate and the absolute value of the rotation angle.

| MNIST | CNN | H-Net | G-CNN | DREN |
|---|---|---|---|---|
| Corr. coeff. | 0.28 | 0.12 | 0.18 | 0.10 |

| CIFAR-10 | ResNet | G-ResNet | | ImageNet | ResNet | OR-ResNet |
|---|---|---|---|---|---|---|
| Corr. coeff. | 0.07 | 0.33 | | Corr. coeff. | 0.06 | 0.05 |

