# OpenReview forum: "Robustness of Rotation-Equivariant Networks to Adversarial Perturbations"
_ICLR.cc/2018/Workshop — Reject_

### Official Review · AnonReviewer2 · 2018-03-02

**Rating:** 5
**Confidence:** 4

**Review:**

This paper investigates the robustness of several rotation-equivariant architectures to adversarial perturbations of a geometric kind. The investigated architectures are CNN, H-Net, G-CNN, DREN and OR-Net. It is found that all of these except H-Net show increased robustness to adversarial samples created by rotating or translating the input. At the same time H-Net does well against spatial transformer-based adversarial attacks.

It seems obvious that networks designed to be invariant to rotations would not suffer from rotation-based adversarial samples. However, because most networks tested only target equivariance to discretized rotations, the results are actually quite interesting / surprising.

The near-term practical significance of this work is limited, because an attacker is not limited to geometrical perturbations when generating adversarial samples, and equivariant networks are still vulnerable to vanilla adversarial samples. Still, this work may have something important to say, which is that perhaps adversarial samples can be thwarted if we can somehow (probably by some kind of unsupervised learning) learn the symmetries in the data, including higher-level non-geometric and local symmetries. In my opinion the paper could be much stronger if it spent some time discussing such implications of the findings.

After looking at the DREN paper, I became a bit confused about the results in Table 1. I could be wrong, but it seems to me like the DREN paper is basically just explaining the 90-degree rotation version of a G-CNN in different notation. The cycle layer is called “first layer group convolution” in the G-CNN paper, and the isotonic layer is just a “group convolution for higher layers”. There is no decycle layer in the G-CNN paper, but it seems like its just equivalent to a sum-pooling plus planar convolution. That the two are equivalent is also supported by [1], which shows that any equivariant network is equivalent to a G-CNN. (This does not apply to OR-Net, because they use finer discretizations of the group convolution, or to H-Net, because they aim to be equivariant to continuous rotations.)

If this is correct, then the differences in the results between G-CNN and DREN must be due to a difference in architecture, model size, initialization, optimization, or something else that should be kept constant. Indeed it's not clear we even need to compare to DREN if this is correct.

Pros/cons:
+ Somewhat surprising empirical findings
+ Potential implications for future research on adversarial samples
- Not useful “in the wild” right now (the authors did not claim it is)
- Some confusing results regarding DREN / G-CNN

[1] Kondor, Risi, and Shubhendu Trivedi. “On the Generalization of Equivariance and Convolution in Neural Networks to the Action of Compact Groups.”

---

> ### Public Comment · ~Beranger_Dumont1 · 2018-05-16
> **feedback on review**
>
> We thank the anonymous reviewer for his constructive and detailed review.
>
> We also agree that the results regarding DREN / G-CNN were confusing. Indeed, DRENs should be equivalent to G-CNNs with the group p4, up to differences in the proposed architecture and learning procedure (we thank the reviewer for pointing us to the paper by Kondor et al.). We have since noticed that the architecture considered for DRENs was mistakenly different from the one of the other networks considered for MNIST in our study. This issue will be fixed in a future version of the manuscript.
>
> Kind regards,
> Beranger Dumont, Simona Maggio & Pablo Montalvo

---

### Official Review · AnonReviewer3 · 2018-03-07

**Rating:** 4
**Confidence:** 4

**Review:**

This paper investigates the robustness of CNNs against geometric adversarial perturbations (such as small translations and rotations). In particular, different CNN architectures designed for rotation-equivariance are studied.

The short paper is presented clearly.
However, the novelty is limited. All neural network architectures considered in the paper are based on previous methods. The contribution of this paper is mainly limited to testing the existing networks using adversarial examples.
Also, this paper considers only small translation and rotation. The topic addressed by this paper is also limited.

Overal, the novelty and significance of this paper seem insufficient.

---

### Public Comment · ~Beranger_Dumont1 · 2018-02-20
**Updated version on arXiv**

Dear reviewers,

A new version of the manuscript just appeared on arXiv: see arXiv:1802.06627 (https://arxiv.org/abs/1802.06627).
The results are practically unchanged, but we note the following differences with respect to the version submitted to OpenReview:
- the presentation of the results has been improved;
- new information on the experimental setup has been added;
- the results in the Appendix have been expanded;
- and finally some new references have been added.

Unfortunately, we cannot update the PDF version on OpenReview right now, but we will update and replace it with the new version from arXiv:1802.06627 as soon as the review period is over.
 We apologize to all reviewers who might already have read the (now out-dated) version on OpenReview, and encourage future reviewers to refer to the improved version at https://arxiv.org/abs/1802.06627 as it might clarify some previously unclear points.

Kind regards,
Beranger Dumont, Simona Maggio & Pablo Montalvo

---

### Decision · Program_Chairs · 2018-03-20
**ICLR 2018 Workshop Acceptance Decision**

**Decision:**

Reject

**Comment:**

Based on the reviews, this paper has not been accepted for presentation at the ICLR workshop. However, the conversation and updates can continue to appear here on OpenReview.